# ELK1/MTOR/S6K1 Pathway Contributes to Acquired Resistance to Gefitinib in Non-Small Cell Lung Cancer

**DOI:** 10.3390/ijms25042382

**Published:** 2024-02-17

**Authors:** Lei Zhao, Yifang Wang, Xin Sun, Xiujuan Zhang, Nicole Simone, Jun He

**Affiliations:** 1Department of Pathology and Genomic Medicine, Sidney Kimmel Cancer Center, Thomas Jefferson University, Philadelphia, PA 19107, USA; lei.zhao@jefferson.edu (L.Z.); yifang.wang@students.jefferson.edu (Y.W.); xin.sun@jefferson.edu (X.S.); 2Department of Medicine, Center for Translational Medicine, Thomas Jefferson University, Philadelphia, PA 19107, USA; xiujuan.zhang@jefferson.edu; 3Department of Radiation Oncology, Sidney Kimmel Cancer Center, Thomas Jefferson University, Philadelphia, PA 19107, USA; nicole.simone@jefferson.edu

**Keywords:** EGFR-TKI, gefitinib, S6K1, MTOR, ELK1, resistance

## Abstract

The development of acquired resistance to small molecule tyrosine kinase inhibitors (TKIs) targeting epidermal growth factor receptor (EGFR) signaling has hindered their efficacy in treating non-small cell lung cancer (NSCLC) patients. Our previous study showed that constitutive activation of the 70 kDa ribosomal protein S6 kinase 1 (S6K1) contributes to the acquired resistance to EGFR-TKIs in NSCLC cell lines and xenograft tumors in nude mice. However, the regulatory mechanisms underlying S6K1 constitutive activation in TKI-resistant cancer cells have not yet been explored. In this study, we recapitulated this finding by taking advantage of a gefitinib-resistant patient-derived xenograft (PDX) model established through a number of passages in mice treated with increasing doses of gefitinib. The dissociated primary cells from the resistant PDX tumors (PDX-R) displayed higher levels of phosphor-S6K1 expression and were resistant to gefitinib compared to cells from passage-matched parental PDX tumors (PDX-P). Both genetic and pharmacological inhibition of S6K1 increased sensitivity to gefitinib in PDX-R cells. In addition, both total and phosphorylated mechanistic target of rapamycin kinase (MTOR) levels were upregulated in PDX-R and gefitinib-resistant PC9G cells. Knockdown of MTOR by siRNA decreased the expression levels of total and phosphor-S6K1 and increased sensitivity to gefitinib in PDX-R and PC9G cells. Moreover, a transcription factor ELK1, which has multiple predicted binding sites on the MTOR promoter, was also upregulated in PDX-R and PC9G cells, while the knockdown of ELK1 led to decreased expression of MTOR and S6K1. The chromatin immunoprecipitation (ChIP)-PCR assay showed the direct binding between ELK1 and the MTOR promoter, and the luciferase reporter assay further indicated that ELK1 could upregulate MTOR expression through tuning up its transcription. Silencing ELK1 via siRNA transfection improved the efficacy of gefitinib in PDX-R and PC9G cells. These results support the notion that activation of ELK1/MTOR/S6K1 signaling contributes to acquired resistance to gefitinib in NSCLC. The findings in this study shed new light on the mechanism for acquired EGFR-TKI resistance and provide potential novel strategies by targeting the ELK1/MTOR/S6K1 pathway.

## 1. Introduction

Lung cancer is the leading cause of cancer-related mortality worldwide. There were an estimated 238,340 new cases and 127,070 cancer-caused deaths in 2023 in the U.S. [1]. Non-small-cell lung cancer (NSCLC) accounts for 85% of all lung cancer cases and is associated with a poor prognosis, with the 5-year overall survival (OS) rate estimated at approximately 15–16% [2,3,4]. Surgical resection, chemotherapy, and radiation therapy represent the traditional therapies for NSCLC [5]. In recent years, targeted therapies have emerged as promising therapeutic strategies and play more and more important roles in NSCLC treatment [6]. Among the targeted therapy options, small molecule tyrosine kinase inhibitors (TKIs) targeting epidermal growth factor receptor (EGFR) signaling have shown significant effectiveness in the treatment of NSCLC patients with tumors harboring specific genetic alterations, such as exon 19 deletions and exon 21 L858R point mutations [7]. These mutations result in the constitutive activation of the EGFR pathway, which promotes cancer cell growth and survival; the EGFR-TKIs exert functions by inhibiting the activity of the EGFR tyrosine kinase, thereby blocking this signaling pathway to inhibit tumor growth [7,8]. The EGFR-TKIs have been shown to significantly extend the progression-free survival (PFS) and OS of patients with EGFR mutation-positive NSCLC [9,10]. However, despite the initial response, many patients eventually develop acquired resistance to EGFR-TKIs, leading to tumor relapse, treatment failure, and shortened survival, which largely limit the application of EGFR-TKIs in NSCLC treatment [8,11]. The mechanisms accounting for the acquired resistance include second EGFR-T790 mutation, activation of alternative signaling pathways, histological transformation, etc. [11,12]. Notably, the mechanisms involved in acquired EGFR-TKI resistance remain largely unknown. Hence, it is crucial to investigate the mechanisms underlying acquired resistance to EGFR-TKIs and identify potential therapeutic targets that can restore tumor sensitivity to EGFR-TKIs to improve the efficacy of EGFR-TKI treatment and enhance patient outcomes.

The 70 kDa ribosomal protein S6 kinase 1 (S6K1), a serine/threonine kinase, plays an important role in regulating cell growth, proliferation, and protein synthesis through phosphorylating multiple downstream targets, such as ribosomal protein S6 [13]. Overexpression of S6K1 has been shown to contribute to tumor development, progression, and poor prognosis in different types of cancers, such as breast, lung, and colorectal cancer [14,15,16]. Several studies indicated that S6K1 mediated cisplatin resistance in ovarian cancer [17] and selumetinib resistance in colorectal cancer [18]. However, the role of S6K1 in acquired EGFR-TKI resistance remains largely unknown. Our previous study showed that constitutive activation of S6K1 contributes to resistance against EGFR-TKIs in NSCLC by facilitating MDM2 phosphorylation and stability [19]. This suggests that targeting S6K1 may improve the efficacy of EGFR-TKIs in resistant NSCLC. Therefore, it is crucial to further understand the role of S6K1 in acquired EGFR-TKI resistance and the regulatory mechanisms that account for its increased activity in NSCLC.

In the current study, we induced resistance to gefitinib in an NSCLC patient-derived xenograft (PDX) model and used this model to investigate the role of S6K1 in acquired EGFR-TKI resistance. We further investigated the upstream regulatory molecules responsible for the excessive activation of S6K1 in gefitinib-resistant tumors and cell lines.

## 2. Results

### 2.1. Development of Gefitinib-Resistant PDX Model In Vivo

We established an NSCLC PDX model in mice with acquired resistance to gefitinib by a number of passages under gefitinib treatment, as shown in Figure 1A. To validate the acquisition of gefitinib resistance, we treated the mice bearing passage-matched parental PDX tumors or resistant PDX tumors with gefitinib at a dosage of 100 mg/kg. The resistant PDX tumors grew significantly faster than the parental PDX tumors with gefitinib treatment (Figure 1B). We then dissociated the primary cells from parental and resistant PDX tumors (named PDX-P and PDX-R, respectively) and measured the response of these cells to gefitinib treatment. As expected, the PDX-R cells showed an increased IC_50_ of gefitinib compared with the PDX-P cells (11.38 µmol/L vs. 2.27 µmol/L, Figure 1C). The PDX-R cells also showed higher cell viability than PDX-P cells when exposed to gefitinib (Figure 1D). Moreover, the PDX-R cells showed a significantly decreased apoptotic rate than the PDX-P cells when exposed to gefitinib (Figure 1E). These results indicated the successful establishment of a gefitinib-resistant PDX model, and the primary cells dissociated from the resistant PDX tumors are more resistant to gefitinib than those dissociated from parental tumors.

### 2.2. Inhibition of S6K1 Improved the Efficacy of Gefitinib in PDX-R Cells

Our previous finding suggests a role of S6K1 in EGFR-TKI resistance, and we, therefore, asked whether S6K1 is also activated in our newly developed gefitinib-resistant PDX model and in primary cells dissociated from the PDX tumors. The PDX-R cells had an increased level of phosphor-S6K1, indicating increased activity of S6K1 in PDX-R cells (Figure 2A). PC9G, a gefitinib-resistant derivative of the PC9 lung adenocarcinoma cell line (Appendix A), also showed increased phosphor-S6K1 levels (Figure 2B). We then treated PDX-R cells with or without PF-4708671, a small molecular inhibitor specifically targeting S6K1 [20]. We found that gefitinib alone slightly reduced cell viability and increased cell apoptosis, while the combination of PF-4708671 and gefitinib dramatically reduced induced cell death, indicating that inhibiting S6K1 activity could improve the efficacy of gefitinib in PDX-R cells (Figure 2C,D). Similarly, PC9G cells treated with a combination of PF-4708671 and gefitinib showed a significantly greater increase in cell apoptosis and a decrease in cell viability compared with cells treated with gefitinib or PF-4708671 alone (Figure 2E). We further knocked down S6K1 through transfection of siRNA oligos specifically targeting S6K1 in PDX-R cells (Figure 2F). The S6K1 siRNA-transfected PDX-R cells exhibited a decrease in the IC_50_ of gefitinib (4.16 µmol/L vs. 0.47 µmol/L, Figure 2G) compared to the scramble-transfected cells. Knockdown of S6K1 in PDX-R cells significantly reduced cell viability (Figure 2H) and increased apoptosis (Figure 2I) compared to the control cells in the presence of gefitinib. Additionally, silencing S6K1 through siRNA transfection in PC9G cells (Figure 2J) also resulted in a decrease in the IC_50_ of gefitinib (20.21 µmol/L vs. 5.37 µmol/L, Figure 2K) and a significantly greater reduction in cell viability (Figure 2L) compared to the control cells upon exposure to gefitinib. These results indicated that increased S6K1 activity contributes to the acquired resistance to gefitinib in PDX tumors and NSCLC cell lines.

### 2.3. Increased MTOR Activity Contributes to S6K1-Mediated Gefitinib Resistance

MTOR is a canonical regulator of S6K1 for protein synthesis and translation, and MTOR has been shown to be involved in EGFR-TKI resistance in various cancer types [21]. We found that both total and phosphor-MTOR were upregulated in PDX-R (Figure 3A) and PC9G (Figure 3B) cells, indicating increased MTOR activity in these cells. Knockdown of MTOR by siRNA transfection decreased total S6K1 and almost abolished phosphorylated S6K1 levels in PDX-R cells (Figure 3C). The effect of MTOR silencing on the sensitivity of gefitinib in PDX-R cells was then evaluated. The cells transfected with siMTOR showed a decreased IC_50_ of gefitinib compared with the control cells (10.14 µmol/L vs. 2.61 µmol/L, Figure 3D). Furthermore, with gefitinib treatment, the MTOR-silenced PDX-R cells displayed a greater decrease in cell viability (Figure 3E) and an increase in apoptosis (Figure 3F) compared to scramble-transfected control cells. We also determined the effects of MTOR knockdown in PC9G cells and found that the IC_50_ of gefitinib was decreased in MTOR-silenced cells compared to control cells (21.88 µmol/L vs. 6.83 µmol/L, Figure 3G,H). When exposed to gefitinib, the MTOR-silenced PC9G cells exhibited significantly decreased cell viability compared to the control cells (Figure 3I). These results suggest that the upregulation of MTOR expression and activity partially accounts for the increased S6K1 activity and gefitinib resistance in PDX-R tumors and PC9G cells.

### 2.4. Upregulation of ELK1 Mediates Gefitinib Resistance through MTOR at the Transcriptional Level

The transcription factor ELK1 is widely involved in tumorigenesis, tumor development, and drug resistance [22,23]. We identified multiple potential ELK1 binding sites in the promoter region of MTOR, as predicted by the JASPAR database (Figure 4A). The mRNA and protein levels of ELK1 were upregulated in PDX-R cells compared with the PDX-P cells (Figure 4B). PC9G cells showed higher mRNA and protein levels than the PC9 cells as well (Figure 4C). The pan-cancer analysis of the TCGA database revealed a positive correlation between ELK1 expression and the expression of MTOR in most cancer types, including lung adenocarcinoma and lung squamous carcinoma (Figure 4D). Knockdown of ELK1 by transfection of ELK1-specific siRNA oligos decreased MTOR and S6K1 protein levels (Figure 4E) and downregulated MTOR mRNA expression (Figure 4F) in PDX-R cells. Similarly, in PC9G cells, silencing of ELK1 also caused decreased MTOR and S6K1 protein levels (Figure 4G) and downregulated MTOR mRNA expression (Figure 4H). Furthermore, the ChIP-PCR assay indicated that ELK1 bound to the promoter of MTOR (Figure 4I). We further conducted a dual-luciferase reporter assay. Overexpression of ELK1 (Figure 4J) significantly increased the luciferase activity of the MTOR reporter in PC9 cells, while silencing ELK1 in PC9G cells resulted in decreased luciferase activity of the MTOR reporter, suggesting that ELK1 directly promotes the transcription of MTOR (Figure 4K). We then investigated whether inhibition of ELK1 enhances the efficacy of gefitinib in PDX-R cells. The silencing of ELK1 resulted in a lower IC_50_ of gefitinib (8.38 µmol/L vs. 1.24 µmol/L, Figure 4L) and a significantly greater decrease in cell viability compared to the control cells in the presence of gefitinib (Figure 4M). Similarly, in PC9G cells, knockdown of ELK1 by siRNA transfection also caused decreased IC_50_ of gefitinib (10.11 µmol/L vs. 3.37 µmol/L, Figure 4N) and reduced cell viability upon exposure to gefitinib (Figure 4O). These findings suggest that increased ELK1 levels lead to the upregulation of MTOR expression by directly promoting its transcription. This, in turn, triggers the activation of S6K1, ultimately contributing to the acquisition of gefitinib resistance.

## 3. Discussion

EGFR-TKIs have become first-line therapies for NSCLC patients with active EGFR mutations, with the overall response rate (ORR) of EGFR-TKIs being around 67% [7,24]. Despite EGFR-TKI treatment resulting in significantly longer progression-free survival (PFS) than canonical chemotherapies, the overall survival did not show a significant difference [25,26]. After the initial response to EGFR-TKI therapy, the majority of patients eventually develop resistance, leading to disease progression and limited improvement in clinical outcomes [25,26]. Extensive research efforts have been dedicated to unraveling the underlying mechanisms responsible for acquired resistance to EGFR-TKIs in NSCLC. For example, the most common mechanism of acquired resistance to EGFR-TKIs in NSCLC is the development of a secondary T790M mutation within the EGFR gene, which impairs the drug’s binding affinity to the receptor, thereby reducing the effectiveness of EGFR-TKI therapy [27]. Other mechanisms include activation of alternative signaling pathways (such as the MET pathway or the HER2 pathway), histological transformation (from NSCLC to SCLC), and acquisition of genetic alterations in downstream signaling molecules (such as KRAS or PIK3CA). However, in some cases, the mechanism for acquired EGFR-TKI resistance is still unclear, which warrants further investigation.

S6K1 is a protein kinase involved in regulating cell growth, protein synthesis, and cell survival [28]. S6K1 is activated by phosphorylation in response to growth factors, nutrients, and other stimuli [28]. Dysregulation of S6K1 signaling has been associated with various types of cancer, including NSCLC, making it a potential target for therapeutic interventions. Qiu et al. showed that treatment with an S6K1-specific inhibitor, PF-4708671, had inhibitory effects on NSCLC tumor growth both in vitro and in vivo [29]. However, the role of S6K1 signaling in EGFR-TKI resistance has been studied little. In our previous study, we showed that phosphorylation levels of S6K1 were correlated with EGFR-TKI resistance and poor prognosis in NSCLC patients. S6K1 signaling was constitutively activated in resistant cancer cells, and MDM2 was a functional effector of S6K1 in mediating EGFR-TKI resistance. The results suggested that inhibition of S6K1 activity may increase the efficacy of EGFR-TKIs in resistant NSCLC [19]. Notably, the study was mainly based on the established EGFR-TKI-resistant NSCLC cell lines and xenograft tumors in mice. PDX models are generated by implanting patient tumor tissue directly into immunodeficient mice, such as the NSG mouse, allowing the tumor to grow and develop in a host organism [30]. PDX models offer advantages over cell lines by recapitulating tumor heterogeneity, preserving tumor characteristics, predicting drug response, providing long-term stability, and offering translational potential for cancer research [30]. In this study, we further investigated the role of S6K1 in gefitinib resistance by taking advantage of an induced gefitinib-resistant PDX model, which was established by continuous administration of gefitinib in vivo. The PDX tumors became partially resistant to gefitinib compared with the passage-matched parental tumors. The phosphorylated S6K1 level was significantly increased in primary cells dissociated from resistant PDX tumors, e.g., PDX-R cells; inhibition of S6K1 by both PF-4708671 treatment and siRNA transfection restored the sensitivity to gefitinib in PDX-R cells, indicating that constitutive activation of S6K1 contributes to the development of EGFR-TKI resistance in PDX tumors. Moreover, these findings were recapitulated in PC9G cells, and the results were consistent with our previous study [19]. In this study, the gefitinib-resistant PDX tumors were able to more closely mimic the clinical relevance and temporal dynamics of drug resistance development, providing a valuable platform for investigating the molecular mechanisms underlying acquired resistance to gefitinib [31]. The results of this study further validated the findings discovered in other cell lines, indicating the important role of S6K1 in acquired resistance to EGFR-TKIs, and S6K1 antagonists may be beneficial in overcoming EGFR-TKI resistance in NSCLC patients.

S6K1 is a downstream effector of the mTOR signaling pathway. In this study, we found that total and phosphorylated MTOR were upregulated in PRX-R and PC9G cells, and increased activity of MTOR contributed to the increased activation of S6K1. The siRNA-mediated silencing of MTOR also led to increased sensitivity to gefitinib in resistant PDX-R and PC9G cells. These results indicated that increased activity of MTOR contributed to the acquired resistance to gefitinib, and inhibition of MTOR activity may reverse gefitinib resistance in lung cancer. The mTOR signaling pathway is a critical signaling pathway involved in regulating cell growth, metabolism, and survival [32]. Dysregulation of the mTOR pathway has been implicated in several diseases, including cancer, neurodegenerative disorders, and metabolic disorders [33]. Unlike S6K1, a number of studies have shown that MTOR signaling is involved in resistance to EGFR-TKIs [21]. For example, Zhang et al. showed that activation of the AKT/mTOR pathway mediated FGFR1-induced acquired resistance against gefitinib in NSCLC [34]. Additionally, mTOR activation through an Akt-independent pathway was involved in collagen type I-induced EGFR-TKI resistance in lung cancer cells [35]. MTOR inhibition, however, reversed the resistance to EGFR-TKIs in lung cancer [36,37]. Notably, current studies mainly focus on the PI3K/Akt/MTOR signaling pathway and MTOR–autophagy axis; the MTOR–S6K1 axis has rarely been studied. Therefore, the activation of MTOR–S6K1 signaling represents a novel mechanism contributing to acquired resistance to EGFR-TKIs in NSCLC.

ELK1 is a member of the ETS family of transcription factors; it plays a crucial role in regulating gene expression and is involved in various cellular processes such as cell growth, differentiation, and development [38]. ELK1 has been shown to mainly play an oncogenic role in various types of cancer. ELK1 can enhance cell proliferation, survival, and invasion by increasing the expression of genes associated with cell cycle progression and migration and activating the oncogenic signaling pathways, such as the CHOP/DR5 pathway and the GPC3–AS1/GPC3 axis [39,40]. Furthermore, ELK1 has been involved in chemoresistance, including cisplatin and gemcitabine, in different types of cancer [41,42,43]. However, the involvement of ELK1 in EGFR-TKI resistance has been rarely studied. Duan et al. indicated that ELK1 might contribute to gefitinib resistance in NSCLC cells by mediating epithelial–mesenchymal transition (EMT) [44]. In this study, we showed that ELK1 was upregulated in PDX-R and PC9G cells, and inhibition of ELK1 re-sensitized the resistant cells to gefitinib. These results indicate that upregulation of ELK1 contributes to the acquired resistance to EGFR-TKIs in NSCLC, and targeting ELK1 may represent an effective therapeutic strategy for restoring sensitivity to EGFR-TKIs in resistant tumors. ELK1 is a well-known transcription factor for multiple genes, such as spastic paraplegia type 4 (SPG4) and kinesin family member C1 (KIFC1) [45,46]. Its relationship with MTOR, however, has not yet been reported. In this study, we identified that ELK1 could directly bind to the promoter of MTOR, and a dual-luciferase reporter assay indicated that ELK1 could promote MTOR promoter transcription activity. Moreover, the silencing of ELK1 decreased MTOR/S6K1 expression. Collectively, these findings indicate that ELK1 could enhance the activity of MTOR/S6K1 signaling by directly promoting the transcription of MTOR. Notably, a luciferase reporter assay needs to be performed to confirm that ELK1 can directly promote the transcription of MTOR. In addition to direct transcriptional regulation, ELK1 has previously been shown to enhance mTOR phosphorylation by reducing DEPTOR transcription [47]. Hence, ELK1 may regulate MTOR expression and activity through multiple pathways, representing the complexity of gene expression regulation in cells. Furthermore, ELK1 is known to be activated by the RAS/RAF/MEK/ERK signaling pathway in breast epithelial cells [48], and RAS activation is one of the mechanisms for EGFR-TKI resistance in lung cancer [49]. However, whether RAS signaling can upregulate ELK1 expression in the context of EGFR-TKI resistance in NSCLC is unclear. More studies need to be performed to investigate the mechanism(s) accounting for the upregulated ELK1 in the development of EGFR-TKI resistance. Notably, although this study indicated that ELK1/MTOR contributed to the constitutive activation of S6K1 in EGFR-TKI-resistant cells, MTOR-independent mechanisms may also lead to S6K1 activation. For example, phosphoinositide-dependent protein kinase PDK1 has been shown to phosphorylate and activate S6K1 directly [50,51]. On the contrary, serine/threonine protein phosphatase 2A (PP2A), an abundant intracellular serine/threonine (Ser/Thr), has been shown to bind and dephosphorylate S6K1 directly [52]. However, whether these mTOR-independent regulatory mechanisms contribute to S6K1 activation in the scenario of EGFR-TKI resistance is still unknown; further studies are warranted for their roles in acquired EGFR-TKI resistance.

## 4. Materials and Methods

### 4.1. Ethics of the Animal Studies

All animal studies were approved by the Institutional Animal Care and Use Committee (IACUC) of Thomas Jefferson University, Philadelphia, PA, USA (#02410), and all animal care and handling procedures were performed following the Guidelines for the Care and Use of Laboratory Animals.

### 4.2. Establishment of the NSCLC PDX Model

Female NOD.Cg-Prkdcscid Il2rgtm1Wjl/SzJ mice (also known as NSG or NOD Scid gamma) bearing TM00784 (LG1208aF) lung adenocarcinoma were obtained from Jackson Laboratory (Bar Harbor, ME, USA). The PDX model was established through serial passages of the tumor tissues in NSG mice (Jackson Laboratory). Briefly, mice bearing PDX tumors were euthanized, and the tumors were harvested and minced into small pieces that could pass through an 18 G needle. Subsequently, the minced tumor tissues were injected subcutaneously into the flanks of NSG mice to generate a new passage of the PDX tumor.

### 4.3. Development of the Gefitinib-Resistant NSCLC PDX Model In Vivo

Mice bearing the parental NSCLC PDX tumors were orally treated with gefitinib (Selleck, Cat# S1025, Huston, TX, USA) three times per week with a starting dose of 20 mg/kg. As tumors showed signs of regrowth despite continuous gefitinib exposure, tumor tissues were harvested, and the PDX model was serially passaged in subsequent generations with continuous and increasing doses of gefitinib administration. This process was repeated through multiple passages, and the dosage of gefitinib was gradually increased to 100 mg/kg. The whole process took 18 weeks until stable gefitinib resistance was achieved. Passage-matched parental PDX tumors were generated as controls.

To validate the establishment of the resistant PDX model, the resistant tumor tissues and passage-matched parental tumors were subcutaneously injected into the flanks of five NSG mice (Jackson Laboratory). The mice were orally administered with 100 mg/kg gefitinib (Selleckchem, Huston, TX, USA) twice per week for 4 weeks. The long and short dimensions of the tumors were measured every six days. Tumor volumes were calculated using the following formula: volume (mm^3^) = [width^2^ (mm^2^) × length (mm)]/2.

### 4.4. Dissociation of Primary Cells from the PDX Tumor

The primary cells were dissociated from the parental and resistant PDX tumors, as described previously [53]. Briefly, the freshly harvested PDX tumors were finely minced into small fragments in DMEM/F12 media (Gibco, Grand Island, NY, USA) with 2% FBS on ice and enzymatically digested using a cocktail of collagenases (0.17 mg/mL Collagenase Type 1, 0.056 mg/mL Collagenase Type 2, and 0.17 mg/mL Collagenase Type 4, all from Worthington Biochemical, Lakewood, NJ, USA) and 0.025 mg/mL Deoxyribonuclease I (Sigma-Aldrich, St. Louis, MO, USA) in DMEM/F12 media with 2% FBS at 37 °C for 2 h with mild shaking. After incubation, tissue pieces were pipetted vigorously to dissociate the cells; the suspension was then spun down to pellet the primary cells. The cells were re-suspended in 10 mL of RBC lysis buffer (Thermo Scientific, Waltham, MA, USA) at RT for 10 min. Subsequently, the cells were washed twice with PBS and filtered through a 70 μm cell strainer (Corning, Corning, NY, USA). The dissociated primary cells were then cultured in Ham’s F12 medium (Gibco) supplemented with 10% FBS (Gibco), 1% penicillin–streptomycin (Gibco), 1.5 g/L sodium bicarbonate (Sigma), 2.7 g/L glucose, 2.0 mM L-glutamine (Gibco), 0.1 mM non-essential amino acids (Gibco), 0.005 mg/mL insulin (Sigma), 10 ng/mL EGF (Sigma), 0.001 mg/mL transferrin (Sigma), and 500 ng/mL hydrocortisone (Sigma) in a humidified atmosphere containing 5% CO_2_ at 37 °C.

### 4.5. Cell Culture

PC9 and PC9G cells were obtained from the American Type Culture Collection (ATCC, Manassas, VA, USA) and cultured in RPMI 1640 Medium (Gibco) supplemented with 10% FBS in a humidified atmosphere containing 5% CO_2_ at 37 °C.

### 4.6. Transfection of the siRNA Oligos

The dissociated primary cells from the resistant PDX tumors and the PC9G cells were plated in 6-well plates. When the confluence reached 60%, the cells were transfected with 100 nmole/L siRNA oligos specifically targeting ELK1, MTOR, or S6K1 (all SMARTpool siRNAs, Dharmacon, Lafayette, CO, USA) using the jetPRIME transfection reagent (Polyplus, Dover, DE, USA) according to the manufacturer’s instructions. The cells transfected with scramble siRNA (siNC, Dharmacon) were used as controls. Protein and RNA samples were harvested 72 h after transfection.

### 4.7. Measurement of the IC_50_ of Gefitinib in Cells

Cells were transfected with siRNA oligos for 48 h, and then the cells were replated in 96-well plates at 3000 cells/well. The cells were treated with different concentrations of gefitinib for 72 h, and each concentration had four replicate wells. The 3-(4,5-dimethylthiazol-2-yl)-2,5-diphenyltetrazolium bromide (MTT, Sigma) assay was used to measure the viability of the cells in each well. Briefly, the medium was replaced with 90 µL of fresh medium supplemented with 10 µL MTT (5 mg/mL) per well, and the plates were incubated for 2 h. The medium was removed after incubation, and 100 µL of DMSO was added to each well and incubated for 15 min under shaking. Then, the absorbance at 450 nm was measured using a microplate reader. The IC_50_ of gefitinib was then calculated using GraphPad Prism.

### 4.8. Cell Apoptosis Assay

Cell apoptosis was evaluated using the FITC Annexin V Apoptosis Detection Kit with PI (BioLegend, San Diego, CA, USA) according to the manufacturer’s instructions. FACS data were analyzed using the FlowJo software (FlowJo, Berkeley, CA, USA). The apoptotic cells were defined as Annexin V+/PI− cells.

### 4.9. Real-Time PCR (RT-qPCR)

Total RNA samples were isolated from siRNA-transfected cells using the TRIzol Reagent (Thermo Scientific, Waltham, MA, USA) according to the manufacturer’s instructions. Then, 1.0 μg of total RNA was reverse transcribed into cDNA using the iScript cDNA Synthesis Kit (Bio-Rad, Hercules, CA, USA) according to the manufacturer’s instructions. The mRNA expression levels of ELK1, MTOR, and S6K1 were measured using the SsoAdvanced Universal SYBR Green Supermix (Bio-Rad) on the QuantStudio 3 Real-Time PCR System (Applied Biosystems, Carlsbad, CA, USA) according to the manufacturer’s instructions. Expression levels of ELK1 and MTOR were quantified using the 2^ΔΔCt^ method. The 18S rRNA was used as the internal control. The primer sequences for the qRT-PCR were as follows: MTOR forward 5′-CTCTGGCATGAGATGTGGCA-3’ and reverse 5′-ATGTCCGTTGCTGCCCATAA-3′; ELK1 forward 5′-CCCGTCCGTGGCCTTATTTA-3′ and reverse 5′-CTCTGCATCCACCAGCTTGA-3′; 18S forward 5′-CGCCGCTAGAGGTGAAATTCTT-3′ and reverse 5′-CAGTCGGCATCGTTTATGGTC-3′.

### 4.10. Chromatin Immunoprecipitation (ChIP) Assay

To investigate if ELK1 directly binds to the promoter of MTOR, we conducted a ChIP assay using the SimpleChIP Enzymatic Chromatin IP Kit (Cell Signaling, Danvers, MA, USA), following the manufacturer’s instructions. Cells growing in 15 cm plates at 80–90% confluence were cross-linked with 1% formaldehyde for 10 min at room temperature. The cross-linking reaction was stopped by adding glycine, and the cells were collected and subjected to sonication using a Q800R3 sonicator (Qsonica, Newtown, CT, USA) with a 10 s sonication followed by a 10 s pause, repeated for 5 min at 30% amplitude. The immunoprecipitation step used 2 μg of ChIP-grade rabbit anti-human ELK1 antibody (ab32106, Abcam, Cambridge, MA, USA) per reaction to pull down ELK1-bound genomic DNA fragments, while 2 μg of normal rabbit IgG (included in the kit) was used as a negative control. The purified DNA fragments were employed as templates for PCR amplification using the DreamTaq Green PCR Master Mix (Thermo Scientific) following the manufacturer’s instructions. Subsequently, the PCR products were resolved on a 2% agarose gel, and the intensity of the resulting bands was quantified using the ImageLab software (Bio-Rad). The fold changes in signal intensity were then calculated. The primers for the ChIP assay were as follows: forward 5′-ACCGGACTCCTTGAGTTCAC-3′ and reverse 5′-CCACTCGGGAGAACCAATCG-3′.

### 4.11. Western Blotting

The whole-cell lysates were harvested in the RIPA buffer supplemented with protease and phosphatase inhibitors (Thermo Scientific). The protein concentration was determined using a BCA assay (Pierce, Rockford, IL, USA) according to the manufacturer’s instructions. The cell lysates were separated using 10% SDS-PAGE gels, and the proteins were then transferred to polyvinylidene fluoride (PVDF) membranes (Millipore, Billerica, MA, USA). The membranes were blocked with 5% non-fat milk in PBST buffer for 1 h at room temperature and then incubated with the indicated primary antibodies overnight at 4 °C. After washing with PBST, the membranes were incubated with goat anti-rabbit or anti-mouse secondary antibodies (1:5000, Thermo Scientific) for 2 h at room temperature. The enhanced chemiluminescence (ECL, Pierce) was then applied, and the protein bands were imaged using a ChemiDoc Imaging System (Bio-Rad), which was quantified using the Image Lab software v6.0 (Bio-Rad). The primary antibodies used in this study included rabbit anti-ELK1 (1:1000, Abcam), rabbit anti-total MTOR (1:1000, Cell Signaling, Danvers, MA, USA), rabbit anti-phosphor-MTOR (1:1000, Cell Signaling), rabbit anti-total S6K1 (1:1000, Cell Signaling), rabbit anti-phosphor-S6K1 (1:1000, Cell Signaling), and mouse anti-β-actin (1:1000, Santa Cruz Biotech, Santa Cruz, CA, USA).

### 4.12. Dual-Luciferase Reporter Assay

Single-stranded DNA from the −318 to −204 region of the MTOR promoter, which contains five predicted ELK1 binding sites in tandem, was synthesized by IDT (Coralville, IA, USA). After annealing, double-stranded DNA was inserted between the KpnI and XhoI sites into a pGL4.17 vector (Promega, Madison, WI, USA) to construct an MTOR promoter luciferase reporter (pGL-MTOR). To observe the effect of ELK1 overexpression on the luciferase reporter activity, we plated PC9 cells in 24-well plates. Subsequently, the cells were transfected with 0.3 µg pGL-MTOR plus 0.1 µg pRL-TK (Promega, internal control), along with 0.4 µg of the pCGN empty vector or pCGN-ELK1 plasmid using the jetPRIME transfection reagent (Polyplus) following the manufacturer’s instructions. To observe the effect of ELK1 silencing on MTOR reporter luciferase activity, we plated PC9G cells in 24-well plates. Subsequently, the cells were transfected with 0.3 µg of the pGL-MTOR plus 0.1 µg pRL-TK (Promega, internal control), along with control siRNA or ELK1 siRNA (at a final concentration of 100 nM) using the jetPRIME transfection reagent (Polyplus). For both assays, cells were incubated for 48 h after co-transfection. The luciferase activity was then measured using the dual-luciferase assay kit (Promega) according to the manufacturer’s instructions. Firefly luciferase activity was normalized to Renilla luciferase activity to indicate the transcription activity of the MTOR promoter reporter.

### 4.13. Statistical Analysis

Data were analyzed using GraphPad Prism software (version 8.0). An unpaired *t*-test was used to compare two groups, while a one-way analysis of variance (ANOVA) followed by a post hoc analysis was used to compare data among three or more groups. Statistical significance was set at *p* < 0.05.

## 5. Conclusions

In summary, this study revealed the promoting role of constitutive S6K1 activation in the acquisition of gefitinib resistance by utilizing a gefitinib-resistant PDX tumor model. We also identified ELK1 as an upstream factor of the canonical MTOR–S6K1 axis and found that overactivation of the ELK1/MTOR/S6K1 pathway contributes to the development of gefitinib resistance in NSCLC. Targeting this pathway represents a potentially effective therapeutic strategy for restoring the sensitivity to EGFR-TKIs in resistant NSCLC tumors, ultimately improving the therapeutic effectiveness and outcome of these patients.

## Figures and Tables

**Figure 1 ijms-25-02382-f001:**
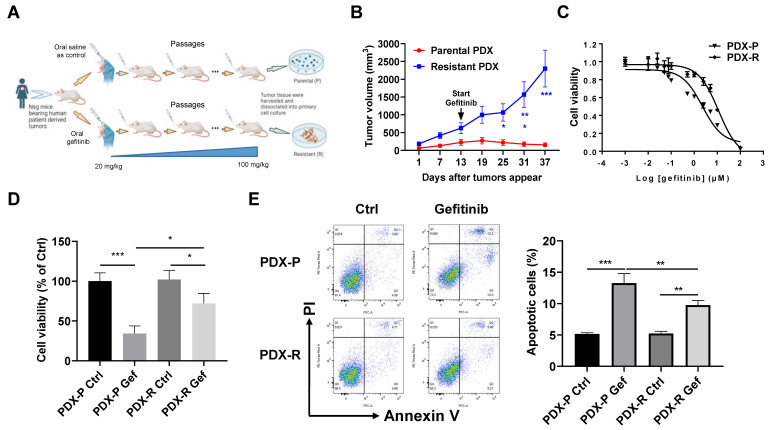
Induction of the acquired gefitinib-resistant PDX model in vivo. (**A**) A schematic figure illustrating the process of inducing the acquired gefitinib-resistant PDX model in vivo. (**B**) The mice bearing passage-matched parental or resistant PDX tumors were orally administrated with 100 mg/kg gefitinib twice per week for 4 weeks. The tumor volume was measured every 6 days. The growth curve of each group was plotted. (**C**–**E**) The PDX tumors were digested with a collagenase cocktail to obtain primary cells in single-cell suspension. (**C**) The primary cells dissociated from parental or resistant PDX tumors were plated in 96-well plates at 3000 cells/well and then treated with different concentrations of gefitinib for 72 h. The cell viability of each well was measured using an MTT assay, and the IC_50_ of gefitinib was then calculated based on the MTT results. (**D**) Cells were plated in 6-well plates and treated with DMSO or 5 µmol/L gefitinib for 72 h. Cell viability was evaluated by counting the cell number. (**E**) Cells were treated with DMSO or 5 µmol/L gefitinib for 72 h. The apoptotic rate of the cells was measured using the Annexin V/PI double-staining flow cytometer. * *p* < 0.05; ** *p* < 0.01; *** *p* < 0.001.

**Figure 2 ijms-25-02382-f002:**
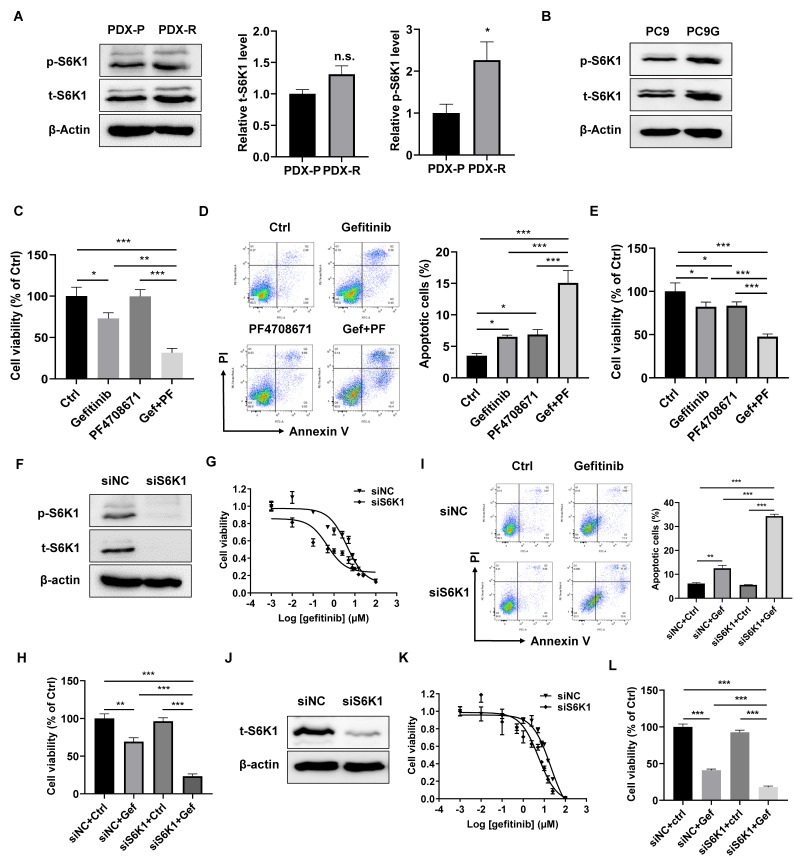
S6K1 contributes to the acquired resistance of lung cancer cells to gefitinib. (**A**) Protein expression of total and phosphorylated S6K1 was measured in the PDX-P and PDX-R cells using immunoblotting. The intensity of the bands was quantified. (**B**) Protein expression of total and phosphorylated S6K1 was measured in PC9G cells using immunoblotting. (**C**,**D**) The PDX-R primary cells in 6-well plates were exposed to DMSO, 5 µmol/L gefitinib, 5 µmol/L PF-4708671, or gefitinib plus PF-4708671 for 72 h. (**C**) Cell viability was evaluated by counting the cell number per well. (**D**) Cell apoptosis was measured using the Annexin V/PI double-staining flow cytometer. (**E**) PC9G cells were treated with DMSO, 5 µmol/L gefitinib, 5 µmol/L PF-4708671, or gefitinib plus PF-4708671 for 72 h. Cell viability was evaluated by counting the cell number. (**F**–**I**) PDX-R cells were transfected with 100 nmole/L S6K1-specific siRNA to silence S6K1. Cells transfected with 100 nmole/L scramble siRNA (siNC) were used as controls. (**F**) The immunoblot shows the depletion of S6K1 in the PDX-R cells. (**G**) The 72 h IC_50_ of gefitinib in the control and S6K1-depleted PDX-R cells was evaluated using the MTT assay. (**H**) The cells were exposed to DMSO or 5 µmol/L gefitinib for 72 h. Cell viability was measured by counting the cell number. (**I**) Cell apoptosis was measured using flow cytometry, as described above. (**J**–**L**) PC9G cells were transfected with 100 nmole/L S6K1-specific siRNA or scramble siRNA. (**J**) Depletion of S6K1 was validated using immunoblotting. (**K**) The 72 h IC_50_ of gefitinib in the control and S6K1-depleted cells was evaluated using the MTT assay. (**L**) The cells were exposed to DMSO or 5 µmol/L gefitinib for 72 h. Cell viability was measured by counting the cell number. * *p* < 0.05; ** *p* < 0.01; *** *p* < 0.001.

**Figure 3 ijms-25-02382-f003:**
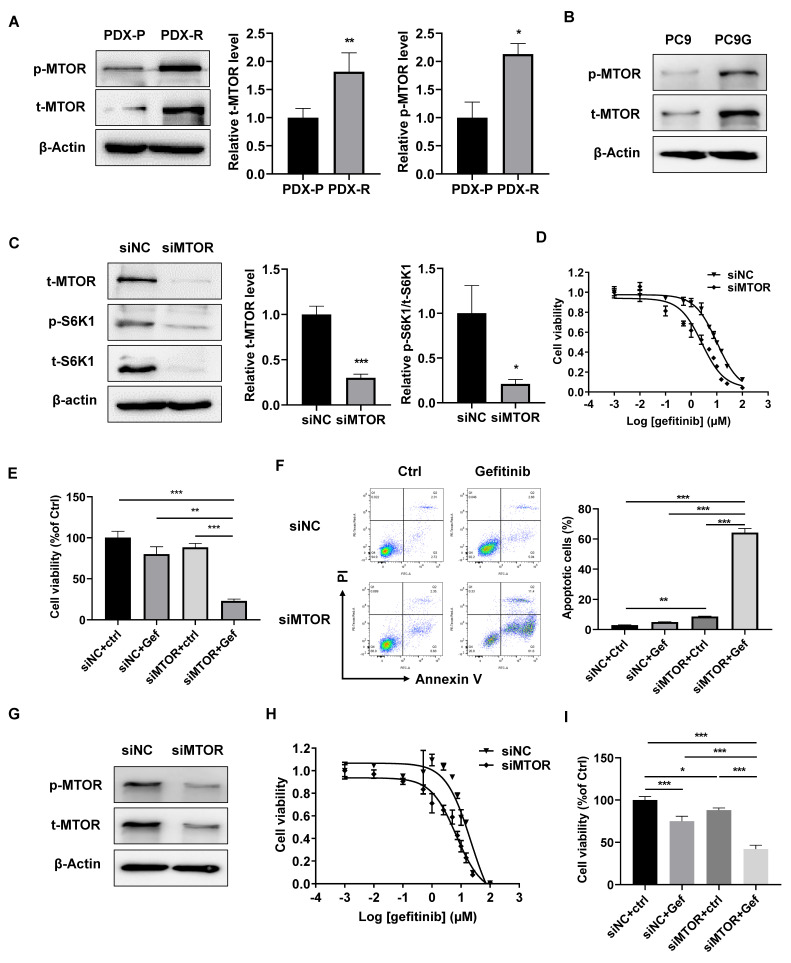
Increased MTOR contributes to S6K1 activation and gefitinib resistance. (**A**) Protein expression of total and phosphorylated MTOR was measured in the PDX-R cells using immunoblotting. The intensity of the bands was quantified. (**B**) Protein expression of total and phosphorylated MTOR were measured in PC9G cells using immunoblotting. (**C**–**F**) PDX-R cells were transfected with 100 nmole/L MTOR-specific siRNA or scramble siRNA. (**C**) The immunoblot shows the depletion of MTOR and the corresponding downregulation of S6K1 in the PDX-R cells. (**D**) The 72 h IC_50_ of gefitinib in the control and MTOR-silenced PDX-R cells was evaluated using the MTT assay. (**E**) The cells were exposed to DMSO or 5 µmol/L gefitinib for 72 h. Cell viability was measured by counting the cell number. (**F**) After exposure to DMSO or 5 µmol/L gefitinib for 72 h, cell apoptosis was measured by flow cytometry, as described above. (**G**–**I**) PC9G cells were transfected with 100 nmole/L siMTOR or scramble siRNA. (**G**) Depletion for MTOR was validated using immunoblotting. (**H**) The 72 h IC_50_ of gefitinib in the control and MTOR-depleted cells was evaluated using the MTT assay. (**I**) The cells were exposed to DMSO or 5 µmol/L gefitinib for 72 h. Cell viability was measured by counting the cell number. * *p* < 0.05; ** *p* < 0.01; *** *p* < 0.001.

**Figure 4 ijms-25-02382-f004:**
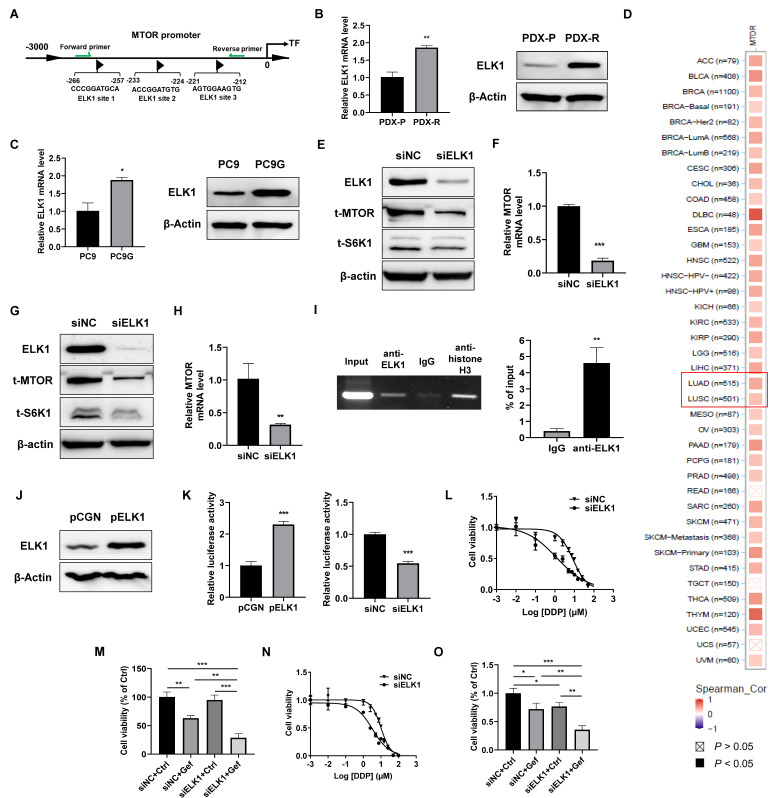
ELK1 directly regulates the transcription of MTOR. (**A**) A schematic figure illustrating the multiple ELK1 binding sites in the MTOR protomer predicted by the JASPAR database. The green arrows indicate the location of the primers for the ChIP assay. (**B**) The mRNA and protein levels of ELK1 in PDX-P and PDX-R cells were measured using qRT-PCR and immunoblotting, respectively. (**C**) The mRNA and protein levels of ELK1 in PC9 and PC9G cells were evaluated using qRT-PCR and immunoblotting, respectively. (**D**) A pan-cancer analysis for the correlation between ELK1 and MTOR expression across different cancer types in the TCGA database. In most cancer types, ELK1 expression is positively correlated with the expression of MTOR. The red rectangle indicates the two types of lung cancers. (**E**,**F**) ELK1 was downregulated in PDX-R cells through transfecting 100 nmole/L siRNA specifically targeting ELK1. Cells transfected with the same amount of scramble siRNA were used as controls. (**E**) The protein expression of ELK1, MTOR, and S6K1 was measured using immunoblotting. (**F**) The mRNA level of MTOR was measured using qRT-PCR. (**G**,**H**) PC9G cells were transfected with siNC or siELK1 to silence ELK1. (**G**) The protein expression of ELK1, MTOR, and S6K1 was measured using immunoblotting. (**H**) The mRNA level of MTOR was measured using qRT-PCR. (**I**) The ChIP-PCR assay was performed to investigate the binding of ELK1 to the MTOR promoter in PC9G cells, as described in the Section 4. The chromatin was immunoprecipitated with an anti-ELK1 antibody and a control IgG. The rabbit anti-histone H3 was used as a positive control. The binding of ELK1 to the MTOR promoter was analyzed using PCR with primers flanking the predicted tandem binding sites shown in (**A**). The PCR products were resolved in 2% agarose gel electrophoresis. The signal of ELK1 and IgG pulldown genomic DNA relative to that of the input was calculated. (**J**) PC9 cells were transfected with the pCGN-ELK1 plasmid or an empty pCGN vector for 48 h. ELK1 protein expression was measured using an immunoblotting assay. (**K**) The effect of ELK1 overexpression or silencing on MTOR promoter reporter luciferase activity was assessed using a dual-luciferase reporter assay. Left: PC9 cells were transfected with pCGN or pCGN-ELK1 together with the pGL-MTOR reporter and pRT-TK plasmids. Right: PC9G cells were transfected with siNC or siELK1 plus the pGL-MTOR reporter and pRT-TK plasmids. For both studies, 48 h after the co-transfection, MTOR reporter luciferase activity was measured using a dual-luciferase reporter assay. (**L**,**M**) ELK1 was silenced in PDX-R cells through siRNA transfection. (**L**) The 72 h IC_50_ of gefitinib in the control and ELK1-silenced cells was evaluated using the MTT assay. (**M**) The cells were exposed to DMSO or 5 µmol/L gefitinib for 72 h. Cell viability was measured by counting the cell number. (**N**,**O**) ELK1 was downregulated in PC9G cells through siRNA transfection. (**N**) The 72 h IC_50_ of gefitinib in the control and ELK1-silenced cells was evaluated using the MTT assay. (**O**) The cells were exposed to DMSO or 5 µmol/L gefitinib for 72 h. Cell viability was measured by counting the cell number. * *p* < 0.05; ** *p* < 0.01; *** *p* < 0.001.

## Data Availability

The original contributions presented in the study are included in the article/Appendix A; further inquiries can be directed to the corresponding author/s.

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
