# Peer review of "ELK1/MTOR/S6K1 Pathway Contributes to Acquired Resistance to Gefitinib in Non-Small Cell Lung Cancer"

_ijms, 2024, doi:10.3390/ijms25042382_

Round 1
Reviewer 1 Report
Comments and Suggestions for Authors
In the manuscript entitled “ELK1/MTOR/S6K1 pathway contributes to acquired resistance to gefitinib in non-small cell lung cancer”, the authors used several approaches to support the hypothesis that ELK1/MTOR/S6K1 pathway contributes to acquired resistance in NSCLC.
The study is interesting, relevant, well conducted, and the manuscript is well written.
There are still minor concerns:
1. In the lines 113/114 the words “therefore” and “asked” are merged.
2. The same for “In” and “our” the line 114.
3. There are few more of these merged words. Please correct.
4. This reviewer suggests the inclusion of a figure summarizing the proposed pathway.
Thus, this reviewer RECOMMENDS the publication of the manuscript in IJMS, after the authors address these minor concerns.
Reviewer 2 Report
Comments and Suggestions for Authors
In the present article, Zhao et al built upon their previous finding showing constitutional activation of S6K1 contributing to EGFR-TKI resistance in NSCLC to identify the overactivation of the ELK1/mTOR/S6K1 axis as the underlying mechanism. They used genetic inhibition of ELK, mTOR, and S6K1 and pharmacological inhibition of S6K1 in two gefitinib-resistant cell lines including one isolated from a gefitinib-resistant PDX model to validate their hypothesis. Collectively, they identified ELK1 as an upstream factor of the canonical mTOR/S6K1 axis and demonstrated that hyperactivation of this signaling pathway contributes to gefitinib resistance in NSCLC. Overall, this is an interesting study with a logical flow and the results obtained from the experiments support the conclusion. However, some concerns should be addressed.
1. ELK-1 is known to be activated by the RAS/RAF/MEK/ERK signaling pathway (doi: 10.1101/gad.191101) and as the authors mentioned RAS activation is one of the mechanisms behind EGFR-TKI resistance. Similarly, ELK-1 had been previously shown to enhance mTOR phosphorylation via reduced DEPTOR transcription (doi: 10.3389/fcell.2021.633035). Hence, the findings of this paper may also be reached by a simple corollary and may not be a unique or independent mechanism.
2. ELK-1 was previously shown to enhance mTOR phosphorylation indirectly via reduced DEPTOR transcription (see above). hence, the luciferase-reporter assay must be included in the manuscript to confirm that ELK1 can directly promote the transcription of MTOR even though the authors demonstrated direct binding of ELK-1 to the mTOR promoter.
Round 2
Reviewer 2 Report
Comments and Suggestions for Authors
The authors have aptly responded to my concerns, performed the suggested experiment, and revised the manuscript accordingly. I do not have any more comments or concerns.